# THE SOFTMAX BOTTLENECK DOES NOT LIMIT THE PROBABILITIES OF THE MOST LIKELY TOKENS

**Ronen Basri**
Faculty of Mathematics and Computer Science
Weizmann Institute of Science
`ronen.basri@weizmann.ac.il`

**David W. Jacobs**
Department of Computer Science
University of Maryland
`djacobs@cs.umd.edu`

## ABSTRACT

In many popular transformer architectures, an output projection matrix linearly maps lower-dimensional embeddings into a higher-dimensional space of logits. It has been shown that this leads to a *softmax bottleneck* that prevents the production of arbitrary probability distributions. It has been argued that this limits large language models (LLMs) in their ability to express next token probabilities that perfectly align with the statistics of natural language. We focus on the ability of such models to produce accurate probabilities for just the top-$m$ tokens. We provide theoretical bounds that show that even a randomly initialized projection matrix can successfully do this for rather large values of $m$, supported by empirical results on both random and trained matrices. This raises questions about whether the softmax bottleneck significantly limits the capabilities of LLMs. We also derive bounds on the maximum number of probabilities that any trained output projection matrix can specify.

## 1 Introduction

Transformers[25] have had a huge impact on machine learning, forming the basis for most large-scale foundation models. Understanding the virtues and limitations of the components of transformers is, therefore, an important challenge. Recent works have identified the *softmax bottleneck* produced by the final linear layer and softmax operation, which limits the output space of transformers. In this paper, we provide a new analysis of the softmax bottleneck and propose new practical ways to assess the expressiveness of transformers in the context of next token prediction. We show that, under this lens, limitations in the expressiveness of transformers may be less significant than previously thought.

The softmax bottleneck concerns the *output projection matrix* (also called the unembedding or readout matrix), which provides a linear map from relatively low-dimensional embeddings into the higher-dimensional space of a logit for each token. Large Language Models (LLMs) apply softmax to these logits to produce a probability distribution over the next token in a sequence. Work on the softmax bottleneck points out that the output projection matrix (OPM) maps embeddings to a low-dimensional linear subspace in the space of all logits, limiting the possible distributions over the next token. [26, 13, 2, 7, 6] argue that this limitation handicaps the performance of LLMs.

A key question is which probability distributions cannot be represented by LLMs due to this bottleneck, and how significant are they to performance? When LLMs use nucleus sampling, only the most likely tokens affect the next token probability at inference time [11]. When text is generated using methods like beam search [8, 5], it is also the case that low-probability tokens may not affect generation. This suggests that it may be most important for LLMs to represent the probability of the most likely tokens as accurately as possible, with the exact probability of very unlikely tokens mattering less.

For this reason, we consider the following question. Given any arbitrary set of $m$ tokens, does the OPM allow us to produce a probability distribution in which they are the most likely tokens? And given specific probabilities for those $m$ tokens that sum to nearly 1, how closely can we generate those probabilities in an LLM? How do the answers to these questions vary with $m$?

We show that for realistically sized models, there exist OPMs such that any set of $m$ tokens can be assigned any specific probabilities that sum to near 1 for rather large $m$. Moreover, this can be done using a random matrix initialized with i.i.d. Gaussian entries. For example, for the embedding and vocabulary size used in GPT-2, we derive a theoretical lower bound that shows that with probability near one, we can generate any desired probabilities (that sum to nearly 1) for up to any $m \approx 26$ tokens using a random OPM. Experimentally, we show that, in fact, this is possible for $m$ up to about 90. For larger models, like GPT3-175B the lower bound rises to over 400, while for Llama-7B the empirical bound rises to over 1,000. We conjecture, and support with empirical evidence (see Appendix A), that at inference time the probabilities of tokens beyond the top 1,000 will not significantly affect performance when nucleus sampling is used. This further raises questions about whether the softmax bottleneck will affect performance significantly at training time.

We then consider the potential capabilities of trained OPMs. First, we examine the following question. What is the largest $m$ for which there exists an OPM such that, for any collection of $m$ tokens, there exists an embedding that the matrix will map to a distribution in which those are the most probable tokens? We use results on the sign rank of matrices to theoretically bound this $m$, showing that it is significantly higher than even the large values of $m$ produced by random matrices.

Finally, we experimentally examine the trained OPMs for GPT-2, TinyLlama, T5-Large, GPT2-XL, and Llama2. We find that their ability to specify probabilities for random collections of tokens is similar to that of random matrices.

Our primary contribution is to show that although the softmax bottleneck limits the set of probability distributions that an LLM can produce, it does not do so in a way that significantly restricts its ability to accurately determine the probability of the most likely next tokens in a sequence.

## 2 Prior Work

[26] provided the initial discussion of the *softmax bottleneck*. They show that models that linearly map a low-dimensional embedding space to a high-dimensional logit space can only represent a low-dimensional linear subspace in the space of log probabilities of all tokens. They conjecture that in natural language, the space of log probabilities is of high dimension and cannot be captured by this subspace. They propose a mixture of softmaxes to overcome this. [13] provides a detailed analysis of the bottleneck. They propose a softmax function that maps the logits to a non-linear subspace, making it possible to cover a wider range of probability distributions. [2] show further limitations. They point out, for example, that if the words *king, queen, woman, man* form a parallelogram, it will be impossible to assign the highest probabilities to both *woman* and *king*. They address this with a *multi-facet softmax*.

These works suggest that model performance is limited by the effective rank of the OPM. [20] raises questions about the relationship between rank and performance, and suggests that improvements from these methods may be due to other factors, such as implicit regularization. More recently, [7] presents empirical evidence that when the rank of the OPM is reduced, model accuracy drops and training loss increases, along with further analysis.

[6] proposes the Linear-Monotonic-Softmax. They also theoretically analyze the distance between arbitrary distributions and those expressible with OPMs, and measure the ability of models to correctly predict the top ($m = 1$) token. [9] further studies the $m = 1$ case, building on results of [3]. They develop a linear program that can determine whether a trained OPM can select a specific token as the most probable one, and show that in many trained models, virtually all tokens can be selected. We use a similar linear program to evaluate this question for the top-$m$ tokens for $m \geq 1$.

Overall, a number of papers have proposed methods to mitigate the effects of the softmax bottleneck, showing some improvements in performance as a result. At the same time, the work of [20] points out that these approaches may be leading to improvement gains for other reasons. Our work raises further questions about whether the softmax bottleneck is truly limiting performance, especially at inference time. We suggest that it remains an open question whether the softmax bottleneck is a problem in need of a solution.

Our work uses the sign rank of matrices to prove lower bounds on the embedding size needed to represent the probabilities of the top $m$ tokens. Sign rank has surfaced in machine learning in various contexts [1]. Of some relevance is the recent work of [15], which uses the sign rank to highlight the limitations of graph representations using attention layers.

More broadly, transformer architectures have been analyzed from a wide range of perspectives. [27, 28, 12] prove that transformers are universal permutation equivariant sequence-to-sequence approximators over compact domains. [16] analyzes the approximation of sparse attention matrices. Other papers explore the computational power of transformers, particularly with Chain of Thought [18, 10, 21]. Finally, [19] shows how transformers can implement gradient descent to learn in-context tasks at inference.

# 3 Preliminaries

We consider an LLM that, in its penultimate layer, has produced a $d$-dimensional embedding, $\mathbf{x}$, for each token in the context. We consider the output projection matrix, $A$, before and after training. This produces $N$ logits, where $N$ is the vocabulary size, so $A$ is $N \times d$. On initialization, $A$ will be random with entries drawn from an i.i.d. Gaussian distribution with zero mean. For simplicity, we assume a variance of 1, but changing the variance does not affect our results. $\mathbf{x}$ produces logits: $\mathbf{y} = A\mathbf{x}$. Letting $\mathbf{a}_j$ denote the $j$'th row of $A$, we can also say that $y_j = \mathbf{a}_j^T \mathbf{x}$. Softmax then produces probabilities as:

$$p_j = \frac{e^{y_j}}{\sum_{i=1}^{N} e^{y_i}} \tag{1}$$

We assume a temperature of 1 here, but our results easily generalize to arbitrary temperatures in the softmax. Without loss of generality, we assume that we desire the first $m$ tokens to have the highest logits, and denote their corresponding logits as $\mathbf{y}_m$ and probabilities as $\mathbf{p}_m$. We denote the first $m$ rows of $A$ as $A_m$, so that $\mathbf{y}_m = A_m\mathbf{x}$. In the next section, we consider the question: given specific values for $\mathbf{p}_m$ and a random matrix $A$, what is the probability that there exists an embedding, $\mathbf{x}$ that will produce the specified values of $\mathbf{p}_m$, along with $\sum_{j=m+1}^{N} p_j < \delta$ for some small $\delta$.

# 4 Random Matrices

In this section, we consider the initialization of $A$ as a random matrix, $A \sim \mathcal{N}(0, 1)$. We begin in Section 4.1 by considering the probability that there exists an $\mathbf{x}$ so that $y_i > y_j, \forall i, j$ such that $i \leq m < j$. This ensures that the first $m$ logits are the largest, which means that the corresponding tokens have the highest probability. We point out that the probability that such an $\mathbf{x}$ exists is maximized when all $y_i$ are equal and positive, for $i \leq m$. Then in Section 4.2, we show that with a slight variation, we can ensure that the $p_i, i \leq m$, all have any desired probabilities that sum to nearly 1, while $p_j, j > m$ can be made arbitrarily small.

## 4.1 Specifying the top-$m$ logits with equal value

For a random output projection matrix $A$, we derive an approximate lower bound on $m^*$, the maximal number of tokens for which $A$ is nearly guaranteed to produce higher probabilities for any $m$-tuple. Proposition 1 states that for a random OPM, the probability that a random set of $m$ tokens has an embedding that makes them more probable than all other tokens (lhs) is bounded by an exponential of a cumulative Gaussian distribution (rhs). We prove this bound by constructing a simple embedding that provides logits for the top-$m$ tokens (WLOG the first $m$), and then estimating the probability that any other logit will be smaller than the largest of these. Corollary 1 then shows that this bound is maximized (and simplified) when the top-$m$ logits are equal.

**Proposition 1.** *Let $A$ be an $N \times d$ matrix whose entries are drawn from an i.i.d. standard normal distribution. For an arbitrary embedding vector $\mathbf{x}$, we let $\mathbf{y} = A\mathbf{x}$. Let $m < d < N$. Then,*

$$P(\exists \mathbf{x}, s.t.\ (y_{m+1}, ..., y_N < y_1, ..., y_m)) \gtrsim \Phi^{N-m} \left( \frac{\min_{i \in [m]} y_i}{\|\mathbf{y}_m\| \sqrt{v}} \right), \tag{2}$$

*where $v$ is the variance of the elements of $\mathbf{a}_j^T A_m^T (A_m A_m^T)^{-1}$, $m + 1 \leq j \leq N$, and is given approximately by*

$$v \approx \frac{d(d-1)}{(d-m)(d-m-1)(d-m-3)}. \tag{3}$$

with the corollary:

**Corollary 1.**

$$\max_{\mathbf{y}_m} P(\exists \mathbf{x}, s.t.\ \mathbf{y}_m = A_m \mathbf{x}, (y_{m+1}, ..., y_N < y_1, ..., y_m)) \gtrsim \Phi^{N-m} \left( \frac{1}{\sqrt{mv}} \right). \tag{4}$$

*and the maximum is obtained for $\mathbf{y}_m = c\mathbf{1}_m$ for any $c > 0$.*

Here $\mathbf{1}_m = (1, ..., 1)^T \in \mathbb{R}^m$.

*Proof.* To derive our bound, we will consider only embeddings $\mathbf{x}$ that lie in the row space of $A_m$. We denote such an embedding by $\mathbf{x}_\|$. So $\mathbf{x}_\| = A_m^T \mathbf{w}$ for some weight vector $\mathbf{w} \in \mathbb{R}^m$, and therefore

$$\mathbf{y}_m = A_m \mathbf{x}_\| = A_m A_m^T \mathbf{w}, \tag{5}$$

and $\mathbf{w}$ is determined by $\mathbf{w} = (A_m A_m^T)^{-1} \mathbf{y}_m$. $A_m A_m^T$ is invertible with probability 1. Thus, there is a unique $\mathbf{x}_\|$ in the row space of $A_m$ that produces $\mathbf{y}_m$. Next, the bottom $N - m$ logits are given by

$$y_j = \mathbf{a}_j^T \mathbf{x}_\| = \mathbf{a}_j^T A_m^T \mathbf{w} = \mathbf{a}_j^T A_m^T (A_m A_m^T)^{-1} \mathbf{y}_m, \quad m < j \leq N, \tag{6}$$

Next, we determine the distribution of $y_j$ and express the probability that it obtains smaller values than $y_i$ for all $i \in [m]$. Since the elements of $A$ are i.i.d. $\mathcal{N}(0, 1)$, the $m \times m$ matrix $(A_m A_m^T)^{-1}$ follows the Inverse Wishart distribution with $d$ degrees of freedom, denoted $\mathcal{W}_m^{-1}(I, d)$ [17] (Sec. 3.8), whose mean is $\mu = \frac{I_m}{d-m-1}$ ($I_m$ denotes the $m \times m$ identity matrix) and its variance, $V$, is

$$V = \text{Var}(A_m A_m^T)^{-1} = \frac{(d-m+1)I_m + (d-m-1)\mathbf{1}_m\mathbf{1}_m^T}{(d-m)(d-m-1)^2(d-m-3)}, \tag{7}$$

Next, we note that the elements of $\mathbf{a}_j^T A_m^T \in \mathbb{R}^m$ are sums of $d$ normal products, so they distribute roughly as $\mathbf{a}_j^T A_m^T \sim \mathcal{N}(0, d)$. For simplicity, we approximate by assuming these values and the entries of $(A_m A_m^T)^{-1}$ are statistically independent. We experimentally verify that the resulting approximation is accurate in Appendix B. We obtain that the elements of the vector $\mathbf{a}_j^T A_m^T (A_m A_m^T)^{-1} \in \mathbb{R}^m$ have zero mean, and their variance is given by $d(V + \mu^2)$. Plugging in the values for $\mu$ and $V$, we notice that all the entries are i.i.d. with zero mean and variance $v$ given by

$$v = \frac{d(d-1)}{(d-m)(d-m-1)(d-m-3)}. \tag{8}$$

Approximating this distribution with a Gaussian $\mathcal{N}(0, v)$, we obtain

$$y_j \sim \mathcal{N}(0, \|\mathbf{y}_m\|^2 v). \tag{9}$$

This implies that for $\mathbf{x}$ in the row space of $A_m$ such that $\mathbf{y}_m = A_m \mathbf{x}$ we have

$$P(y_{m+1}, ..., y_N \leq \min_{i \in [m]} y_i) \approx \Phi^{N-m} \left( \frac{\min_{i \in [m]} y_i}{\|\mathbf{y}_m\| \sqrt{v}} \right), \tag{10}$$

where $\Phi$ denotes the CDF of the standard normal distribution. Note that $\Phi^{N-m}$ is the CDF of the Gumble distribution with location parameter $\mu_{N-m} = \Phi^{-1}\left(1 - \frac{1}{N-m}\right)$ and scale $\sigma_{N-m} = \Phi^{-1}\left(1 - \frac{1}{(N-m)e}\right) - \Phi^{-1}\left(1 - \frac{1}{N-m}\right)$. The bound in (2) is implied. It can be readily verified that this probability is largest when $y_1 = ... = y_m > 0$, and is invariant to scaling of $\mathbf{y}$, proving (4). $\quad\square$

In Table 1 (Appendix C), we show the value of our lower bound for 22 common models. We see values ranging from $m = 26$ for GPT2 to $m = 418$ for GPT3-175B. This shows that for larger current models, the number of tokens that can be specified is quite high.

Note finally that this analysis restricts $\mathbf{x}$ to lie in the row space of $A_m$, and so it provides only a lower bound on the top-$m$ probability. Using linear programming, we show empirically in Section 6 that adding a component $\mathbf{x}_\perp$, orthogonal to the row space of $A_m$, leads to significantly higher probabilities.

## 4.2 Specifying the top-$m$ probabilities

Suppose now that we want to select $\mathbf{x}$ to exactly specify the relative probability of the top-$m$ tokens. This is more complex, since softmax causes the probability of a token to depend on the logits of all tokens. We first show how to construct an embedding in which the ratios of the top-$m$ probabilities achieve specified values, but the sum of these probabilities, and so their exact values, are not fully specified. Then we show how to scale and adjust this embedding so that the sum of the top $m$ probabilities approaches 1. When these values sum close to one, the probability that such an embedding exists for a random OPM is given by our previous results in Prop. 1.

Formally, we say that $\mathbf{x} \in \mathcal{X}_{p_1,\dots p_m}$ if and only if $\frac{e^{\mathbf{a}_i^T \mathbf{x}}}{e^{\mathbf{a}_m^T \mathbf{x}}} = \frac{p_i}{p_m}$ for $1 \leq i \leq m$, and we want to select $\mathbf{x}$ from this set. Without loss of generality, we always assume that $p_m \leq p_i$ for $i < m$. Further, we suppose that only these top-$m$ tokens should be generated, so we wish to assign low probabilities to all other tokens. We will consider the event that a random matrix can generate probabilities so that the top-$m$ tokens have specific ratios of values while the remaining probabilities collectively sum to $\delta$ or less, for some small $\delta$. Our analysis below derives a bound on the probability of this event, as a function of $m$, using the conditions of Prop. 1, i.e., when the embedding $\mathbf{x}$ is restricted to lie in the row space of $A_m$. As $\delta$ goes to 0 this converges to the bounds in (4). We formalize this as:

**Proposition 2.** *Let $A$ be an $N \times d$ matrix whose entries are drawn from an i.i.d. standard normal distribution. Let $\frac{p_i}{p_m}$ be any specified set of ratios for $i < m$. Given $0 < \delta < 1$ and $0 < \epsilon$, with $\mathbf{y} = A\mathbf{x}$:*

$$P(\exists \mathbf{x} \in \mathcal{X}_{p_1,\dots,p_m} \ s.t. \ (y_{m+1}, \dots, y_N \leq \min_{i \in [m]} y_i(1 - \epsilon)) \gtrapprox \Phi^{N-m}\left(\frac{\min_{i \in [m]} y_i(1 - \epsilon)}{\|\mathbf{y}_m\|\sqrt{v}}\right), \quad (11)$$

*with the further condition on $\mathbf{x}$ that if $y_{m+1}, \dots, y_N \leq \min_{i \in [m]} y_i(1 - \epsilon)$, then $\sum_{j=m+1}^N p_j \leq \delta$.*

We have the corollary:

**Corollary 2.** *As $\epsilon$ goes to 0, the conditions of Proposition 2 can be met, and:*

$$\lim_{\epsilon \to 0} \ P(\exists \mathbf{x} \in \mathcal{X}_{p_1,\dots,p_m} \ s.t. \ (y_{m+1}, \dots, y_N \leq \min_{i \in [m]} y_i(1 - \epsilon)) \gtrapprox \Phi^{N-m}\left(\frac{1}{\sqrt{mv}}\right). \quad (12)$$

*Proof.* We create a series of values for $\mathbf{x}$ of increasing scale. We arrange this so that $\mathbf{x} \in \mathcal{X}_{p_1,\dots,p_m}$, but as the scale increases, the sum of their probabilities approaches 1. Let $\mathbf{x}_s$ denote a choice for $\mathbf{x}$ that is determined by a scale factor, $s$, in a fashion described below. We decompose $\mathbf{x}_s = s\bar{\mathbf{x}}_s$. $\bar{\mathbf{x}}_s$ is chosen to produce the desired ratios of probabilities for $p_1, \dots, p_m$, while its scale is fixed by requiring that $\mathbf{a}_m^T \bar{\mathbf{x}}_s = 1$. We further define $\bar{\mathbf{y}}_s = A_m \bar{\mathbf{x}}_s$. That is, $\bar{\mathbf{y}}_s$ represents the logits that would be produced by the unscaled $\bar{\mathbf{x}}_s$. So the logits for scale $s$ are $\mathbf{y}_s = s\bar{\mathbf{y}}_s$. We denote by $\mathbf{p}_s$ the probabilities produced by applying softmax to $\mathbf{y}_s$. We will refer to the $i$'th elements of $\bar{\mathbf{y}}_s$, $\bar{\mathbf{x}}_s$ and $\mathbf{p}_s$ as: $\bar{\mathbf{y}}_s(i)$, $\bar{\mathbf{x}}_s(i)$ and $\mathbf{p}_s(i)$.

Given these definitions, specified ratios of the top-$m$ probabilities, and $s$, we now show how to derive $\bar{\mathbf{x}}_s$ and $\bar{\mathbf{y}}_s$. For a given value of s, we have:

$$\frac{p_i}{p_m} = \frac{e^{s\bar{\mathbf{y}}_s(i)}}{e^{s\bar{\mathbf{y}}_s(m)}} = \frac{e^{s\bar{\mathbf{y}}_s(i)}}{e^s} \quad (13)$$

since $\bar{\mathbf{y}}_s(m) = 1$. So we have

$$\log\left(\frac{p_i}{p_m}\right) = s(\bar{\mathbf{y}}_s(i) - 1), \quad \text{and} \quad \bar{\mathbf{y}}_s(i) = \frac{1}{s}\log\left(\frac{p_i}{p_m}\right) + 1 \quad (14)$$

for $i < m$. So, given $s$ and the ratios $\frac{p_i}{p_m}$ we can determine the first $m$ values of $\bar{\mathbf{y}}_s$. As $s$ goes to infinity, the $\bar{\mathbf{y}}_s(i)$ become arbitrarily close to 1, while for every value of $s$ the $\bar{\mathbf{y}}_s(i)$ will produce $\mathbf{p}_s(i)$ such that $\frac{\mathbf{p}_s(i)}{\mathbf{p}_s(m)} = \frac{p_i}{p_m}$, for $i \le m$. As noted in the previous section, because $A_m$ has full rank, we can compute $\bar{\mathbf{x}}_s$ so that $A_m\bar{\mathbf{x}}_s = \bar{\mathbf{y}}_s(1:m)$, where $\bar{\mathbf{y}}_s(1:m)$ is the vector of $(\bar{\mathbf{y}}_s(1), ..., \bar{\mathbf{y}}_s(m))^T$.

To summarize, we have defined $\bar{\mathbf{x}}_s$ so that the vector $\mathbf{x} = s\bar{\mathbf{x}}_s$ produces logits that lead to the desired ratio of probabilities for the first $m$ tokens, satisfying the first part of the proposition. As $s$ increases the first $m$ elements of $\bar{\mathbf{y}}_s$ become more uniform. We will see that the sum of probabilities for the remaining tokens shrinks, provided that their corresponding logits are smaller than the first $m$ logits.

We note that $\mathbf{y}_s(j) < \min_{i \in \{1...m\}}(\mathbf{y}_s(i))$ if and only if $\bar{\mathbf{y}}_s(j) < \bar{\mathbf{y}}_m = 1$, that is, that scaling all the logits does not affect their order. So the probability that the first $m$ logits are larger than all remaining ones, with a gap of $\epsilon$ in the unscaled logits, is given by (11). As $s$ goes to infinity, the first $m$ values of $\bar{\mathbf{y}}_s$ become arbitrarily close to 1. So in the limit, the probability is given by (4), which applies when the first $m$ logits are equal to 1. This tells us that the probability that the first $m$ logits are the largest is independent of the ratios of these probabilities for large $s$.

We need to show that, as $s$ becomes large, the collective probability of the top-$m$ tokens approaches 1. The value of $s$ needed to achieve this depends on the gap between the $\max_{j \in \{m+1...N\}} \bar{\mathbf{y}}_s(j)$ and 1. So we will suppose that all $\bar{\mathbf{y}}_s(j) < 1 - \epsilon$, for some small $\epsilon$. As $\epsilon$ goes to zero, the probability that this supposition is true will be given by (11).

Suppose we want $\sum_{m+1}^{N} p_j < \delta$, for some small $\delta$. If we have:

$$\frac{\sum_{j=m+1}^{N} \bar{\mathbf{P}}_s(j)}{\sum_{i=1}^{m} \bar{\mathbf{P}}_s(i)} = \frac{\sum_{j=m+1}^{N} e^{s\bar{\mathbf{y}}_s(j)}}{\sum_{i=1}^{m} e^{s\bar{\mathbf{y}}_s(i)}} < \delta, \tag{15}$$

this ensures that $\sum_{j=m+1}^{N} p_j < \delta$. We have:

$$\frac{\sum_{j=m+1}^{N} e^{s\mathbf{y}_s(j)}}{\sum_{i=1}^{m} e^{s\mathbf{y}_s(i)}} \le \frac{N e^{s(1-\epsilon)}}{m e^s} = \frac{N}{m} e^{-s\epsilon}. \tag{16}$$

We can ensure that Equation (15) holds if we choose $s$ so that:

$$\frac{N}{m} e^{-s\epsilon} < \delta \qquad \text{which implies}: \qquad s \ge -\frac{1}{\epsilon} \log \frac{m\delta}{N}. \tag{17}$$

Therefore, choosing $\bar{\mathbf{x}}_s$ as described above, and $s$ sufficiently large, we ensure that the top-$m$ tokens have probabilities with the desired ratios, while the sum of the remaining ones is arbitrarily small. $\quad\square$

These results show that using a random OPM, if an LLM can specify the top $m$ tokens it can also specify their exact probabilities, if they sum close to 1. When using nucleus sampling during inference, this means that for fairly large values of $m$ the LLM can specify the exact probabilities of all tokens that might be sampled.

# 5 Minimal embedding dimension

In this section, instead of showing lower bounds on performance using random OPMs, we consider the upper bound on the possible performance of trained OPMs. We ask: for what value of $m$ can the best possible OPM specify the top-$m$ next tokens, for any set of $m$ tokens? We formalize this as:

**Problem statement:** Let $m \in [N]$. What is the minimal embedding dimension $d^*$ that allows us to assign maximal probabilities to all subsets of tokens of size $m$?

This tells us the limits of the best possible trained matrix. We start with a simple construction for $m = 1$ that shows that any token can be assigned the highest probability when $d \ge 2$.

**Proposition 3.** *For embedding dimension $d = 2$ and $N \ge 2$ there exists a matrix, A, of dimension $N \times 2$ such that, for any $i$, there exists a 2d vector $\mathbf{x}$ such that $\mathbf{a}_i^T \mathbf{x} > \mathbf{a}_j^T \mathbf{x}, \forall j \ne i$.*

*Proof.* Let the rows of $A$ correspond to distinct points on the unit circle, i.e., $\|\mathbf{a}_i\| = 1$. Then if we set $\mathbf{x} = \mathbf{a}_i$, we have $\mathbf{a}_i^T \mathbf{x} = 1$ and $\mathbf{a}_j^T \mathbf{x} < 1$ for any $j \neq i$. □

We next generalize this result to larger values of $m$. We begin by showing that this problem is tightly related to a sign-rank problem for a matrix that reflects the combinatorial structure of the problem. Given a matrix, $S$, containing values of $+1$ or $-1$, the sign rank of $S$ is defined as the minimum rank of any matrix of the same size as $S$ whose elements have the same sign as the corresponding elements of $S$. This connection will enable us to determine tight upper and lower bounds on $d^*$. For our first proposition, we define a matrix $S \in \{-1, +1\}^{N \times \binom{N}{m}}$, which we call an $\binom{N}{m}$-*indicator matrix*. $S$ is defined such that there are exactly $m$ entries in each column with value $+1$ and $N - m$ entries with value $-1$ and each column $\mathbf{s}_j \in \{-1, +1\}^N$ has positive values corresponding to a unique subset $\pi_j \subset [N]$ of size $|\pi_j| = m$.

**Proposition 4.** *Let $S$ denote an $\binom{N}{m}$-indicator matrix. Then,*

$$|d^* - \operatorname{signrank}(S)| \leq 1.$$

*Proof.* Consider a collection of $\binom{N}{m}$ logit vectors, $\mathbf{y}_1, ..., \mathbf{y}_{\binom{N}{m}} \in \mathbb{R}^N$, such that in each vector $\mathbf{y}_j$, a different subset of size $m$ logits are largest. Arrange these logits in a matrix $Y$ of size $N \times \binom{N}{m}$. We call $Y$ a *complete top-$m$ logit matrix*. Suppose $\operatorname{rank}(Y) = d$, then there exist $N \times d$ and $d \times \binom{N}{m}$ matrices $A$ and $X$ such that $Y = AX$. Here, $A$ serves as an output projection matrix and the columns of $X$ include the respective embeddings. Consider next the set of all complete top-$m$ logit matrices, denoted $\mathcal{Y}$. We therefore ask, what is the minimal rank of $Y \in \mathcal{Y}$?

To relate this question to a sign rank problem, note that for every logit vector $\mathbf{y}_j$ in $Y$, there exists a threshold $t_j$ such that the top-$m$ entries are larger than $t_j$ and the remaining entries are smaller than $t_j$. Therefore, $\mathbf{y}_j - t_j \mathbf{1}_N$ is positive only in the top-$m$ entries, where $\mathbf{1}_N$ is a vector of $N$ ones. Let $T$ be defined through the outer product $T = \mathbf{1}_N^T \left( t_1, ..., t_{\binom{N}{m}} \right)$, then clearly the matrix $Y - T$ has the sign pattern of $S$ (with the corresponding permutation of the columns). By definition, the minimal rank of matrices with this sign pattern is given by the sign rank of $S$. Finally, since $\operatorname{rank}(T) = 1$, then $d^* = \min_{Y \in \mathcal{Y}} \operatorname{rank}(Y)$ cannot differ from the sign rank of S by more than 1. □

Our next proposition uses results by Alon et al. [1] to determine the sign rank of $S$ exactly.

**Proposition 5.** *Assuming $N \geq 3m + 1$, then $\operatorname{signrank}(S) = 2m + 1$.*

*Proof.* We begin by proving that $\operatorname{signrank}(S) \leq 2m + 1$. We will use Lemma 19 from [1]. Given a sign matrix $\tilde{S}$, let $SC(\tilde{S})$ denote the maximum number of sign changes along any of its columns, and let $SC^*(\tilde{S}) = \min SC(M)$ where the minimum is taken over all matrices $M$ obtained from $\tilde{S}$ by a permutation of its rows. Then according to this lemma, $\operatorname{signrank}(\tilde{S}) \leq SC^*(\tilde{S}) + 1$. Applying this lemma to our $\binom{N}{m}$-indicator matrix $S$, we notice that every column of $S$ has up to $2m$ changes of sign. Moreover, already when $N \geq 2m + 1$, there exists a column with exactly $2m$ changes of sign, and such a column exists with any permutation of the rows of $S$. Therefore, $SC^*(S) = 2m$, implying that $\operatorname{signrank}(S) \leq 2m + 1$.

We next prove that $\operatorname{signrank}(S) \geq 2m + 1$. For this, we use the notion of dual sign rank, as defined in [1] and in Claim 15 therein. To match our definitions, we exchange the roles of rows and columns in that claim. We say that a set of rows $C$ is *antipodally shattered* in a sign matrix $\tilde{S}$ if for each vector $\mathbf{v} \in \{\pm 1\}^{|C|}$, either $\mathbf{v}$ or $-\mathbf{v}$ appear as a column in the restriction of $\tilde{S}$ to the rows in $C$. Claim 15 in [1] establishes that the set of rows $C$ is antipodally shattered in $\tilde{S}$ if and only if the sign rank of $S$ is at least $|C|$. The dual sign rank of $\tilde{S}$ is the cardinality of the maximal such set of rows $C$.

Returning to our $\binom{N}{m}$-indicator matrix $S$, we claim that its dual sign rank is exactly $2m + 1$. To see this, notice that with $N \geq 3m + 1$ every set of $2m + 1$ rows is antipodally shattered, while no set of $2m + 2$ rows is antipodally shattered. Specifically, the columns of $S$ include all possible sign configurations with exactly $m$ plus signs, and therefore the restriction of $S$ to the set of rows $C$ with $|C| = 2m + 1$, denoted $S_C$, includes columns with all possible sign configurations with $\leq m$ plus signs. The remaining vectors $\mathbf{v} \in \{\pm 1\}^{2m+1}$ include $> m$ plus signs and $<= m$ minus signs. For

these vectors, $-\mathbf{v} \in -S_C$. Consequently, the union of columns in $S_C$ and $-S_C$ include all vectors in $\{\pm 1\}^{2m+1}$, implying that $C$ is antipodally shattered. On the other hand, with $|C| >= 2m + 2$, there exist vectors in $\{\pm 1\}^{|C|}$ that have both $> m$ plus signs and $> m$ minus signs. Such vectors are neither in $S_C$ nor in $-S_C$, implying that the dual sign rank of $S$ is $2m + 1$. Since the dual sign rank lower bounds the sign rank, we have proved our proposition. $\square$

Propositions 4 and 5 immediately imply the following bounds on $d^*$.

**Corollary 3.** *Assuming $N \geq 3m + 1$, then $2m \leq d^* \leq 2m + 2$.*

In conclusion, Corollary 3 tells us that for a desired $m$, there exists an $N \times d$ matrix with $2m \leq d \leq 2m + 2$ that enables assigning the highest probabilities to any $m$ desired tokens. Note, however, that a higher embedding dimension may be needed to allow prescribing any desired probabilities to the top-$m$ tokens, or even just to assign them any desired order. Our results show that there exist OPMs that significantly exceed the performance of random matrices analyzed earlier. For example, our lower bounds show that a random OPM in a GPT2-sized model can specify roughly the most likely 26 tokens, while our bounds in this section show that there exists an OPM that can specify the top 383 tokens in such a model.

# 6 Experiments

We perform a number of experiments to provide more precise information about the behavior of output projection matrices.[1] In each experiment, we select $m$ random tokens, varying $m$. We then evaluate the ability of the OPM to generate appropriate logits and probabilities for these tokens.

We experiment using the trained OPMs for GPT-2 and GPT2-XL [22], TinyLlama-1.1B [29], T5-Large [23] and Llama2-7B [24], and also with random matrices of the same size as these. GPT-2 has $N = 50,257$ tokens and an embedding size of $d = 768$ andLlama2-7B has $N = 32,000$, $d = 4,096$, while the other models are of intermediate size. We note that the weights of the OPM are tied to the embedding weights in the GPT models and T5, but not in the Llama models.

Our derived lower bound (Section 4.1) assumes each embedding $\mathbf{x}$ lies in the row space of $A_m$. Suppose instead we allow $\mathbf{x}$ to contain a component orthogonal to the space spanned by the rows of $A_m$. We denote by $\mathbf{x}_\parallel$ the component of $\mathbf{x}$ that lies in the row space of $A_m$ and by $\mathbf{x}_\perp$ the component orthogonal to this space, so that $\mathbf{x} = \mathbf{x}_\parallel + \mathbf{x}_\perp$. $\mathbf{x}_\perp$ will not affect the $y_i, i \leq m$. However, it will affect $y_j, j > m$, and can be chosen so that it reduces the largest of these logits. Given either a randomly generated or trained $A$ and a choice of $m$ tokens (without loss of generality, the first $m$ tokens), we use Linear Programming (LP) to find a feasible vector $\mathbf{x}$ such that $A_m\mathbf{x} = \mathbf{y}_m$, where $\mathbf{y}_m$ may be $(1, ..., 1)^T$ when we verify whether we can specify the desired top-$m$ tokens, or specific values when we want to generate specific probabilities. We further require that for all $1 \leq i \leq m < j$, $\mathbf{a}_j^T\mathbf{x} < \mathbf{a}_i^T\mathbf{x}$.

Finally, we measure the ability of matrices to generate precise probabilities for the top-$m$ tokens. We generate the desired probabilities from a uniform distribution between $0.1$ and $1.1$ (this limits the maximum ratio between the largest and smallest probability to 11). In Appendix D we show similar results using probabilities generated by an LLM. We have shown theoretically that, in the limit, as $\epsilon$ and $\delta$ approach 0, the chances that we can generate precise probabilities should approach the probability that we can generate top-$m$ logits with equal probability. In experiments, we use $\epsilon = 0.05, \delta = 0.1$.

In Figure 1(top), for a GPT-2-sized random matrix, we see (dotted purple curve) that our theoretical bound predicts that we can generate any combination of $m \approx 26$ top tokens with probability near 1. In Appendix B we show empirical results that match this closely. Figure 1 also shows that for a TinyLlama-sized OPM with a larger embedding and fewer tokens, the probability is close to 1 for $m \approx 70$ (purple also). We show similar results for three additional models, T5-Large, GPT2-XL and Llama2. Llama2, whose embedding size of 4,096, can generate any $m = 143$ as the top tokens. In Figure 2, we graph this bound for several more models of different sizes.

---

[1]Code to reproduce these experiments is available at `https://github.com/ronenbasri/The-Softmax-Bottleneck-Does-Not-Limit-the-Probabilities-of-the-Most-Likely-Tokens`.

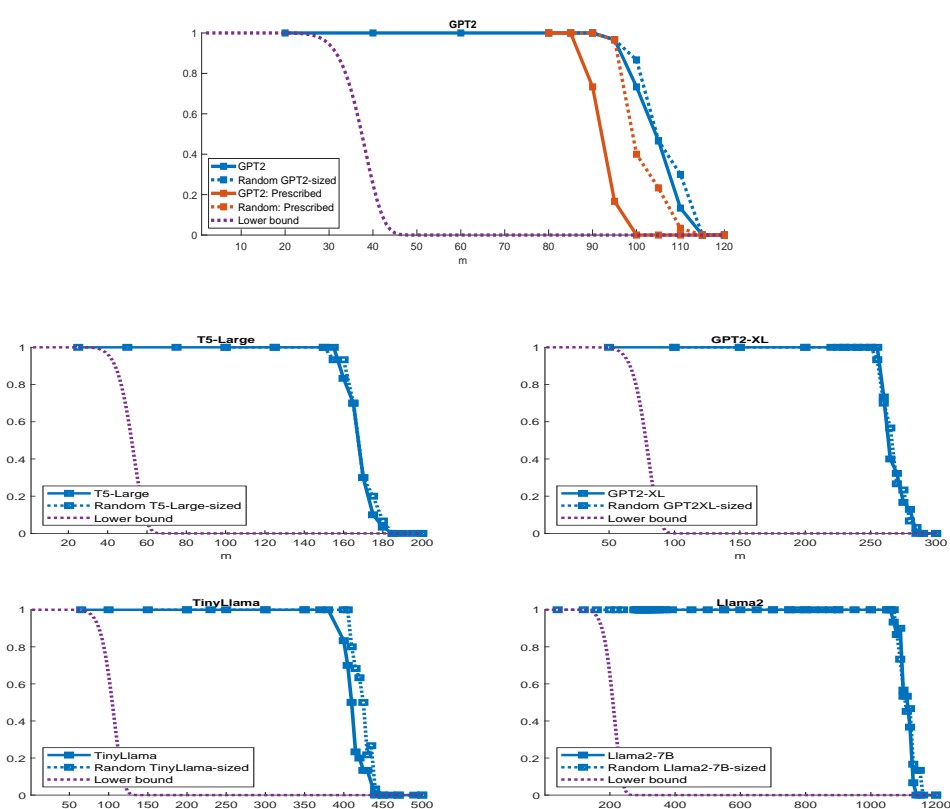

Figure 1: Top: Top-$m$ probability for the GPT-2 and random GPT2-sized ($50257 \times 768$) output projection matrices. Curves show the fraction of times that the matrix satisfied desired constraints for $m$ randomly chosen tokens as $m$ increases. We show our lower bound for a random matrix (4) in dotted purple, along with linear-programming simulations for a random matrix (dotted blue) and the trained GPT-2 (solid blue) matrix. Also in this panel, we show LP simulations with prescribed top-$m$ probabilities (random matrices in dotted red and the trained GPT-2 matrix in solid red). Other: Similarly, for T5-Large, GPT-XL, TinyLlama, and Llama2 and corresponding random matrices of the same size TinyLlama-sized, we show our lower bound (4) in dotted purple along with linear program simulations for random matrices (dotted blue) and the trained matrix (solid blue).

When we incorporate Linear Programming to fully optimize $\mathbf{x}$, we find (Figure 1(top)) that GPT-2 (solid blue) and a same sized random matrix (dotted blue) both have probability close to one of being able to specify $m = 95$ tokens, while TinyLlama, for example, (Figure 1(bottom right)) maintains this up to $m = 400$ (blue), and Llama2 (bottom right) does this for up to $m = 1070$. Using GPT-2, we specify exact probabilities for these tokens (Figure 1(top)). The largest $m$ with probability close to 1 drops slightly, which is expected given our choice of $\epsilon$ and $\delta$ (solid and dotted red). In comparison, using Corollary 3, the best possible trained GPT-2 and TinyLlama-sized matrices can assign the highest probability to all configurations of up to $m = 383$ and $m = 1023$ tokens, respectively.

In an additional experiment (Figure 3), we show the value of $m$ as a function of $d$ and $N$ when LP is applied, for random matrices. With $d \ll N$, we observe that $m \approx d/5$, and $m$ decreases slowly with $N$ as $N$ becomes large. This is an empirical observation, but it provides a simple formula that predicts how large an embedding is needed if one knows how many tokens one needs to represent accurately.

Overall, we conclude that standard OPMs can express fairly large high-probability tuples already at initialization. It seems that training neither improves nor worsens this ability. Many prior works

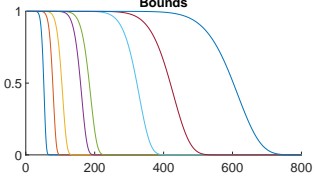

| Model | N | d | m |
|---|---|---|---|
| GPT2 | 50257 | 768 | 26 |
| T5-Large | 32128 | 1024 | 35 |
| GPT2-XL | 50257 | 1600 | 54 |
| TinyLLaMA | 32000 | 2048 | 71 |
| Qwen3 | 152064 | 3584 | 113 |
| LLaMA2-7B | 32000 | 4096 | 143 |
| DeepSeek-V3 | 129280 | 7168 | 230 |
| LLaMA2-70B | 32000 | 8192 | 287 |
| GPT3-175B | 50257 | 12288 | 418 |

Figure 2: For the models in the table (right), we show the probability of assigning the highest probabilities to the top-$m$ logits according to the bound in (4) (left, graphs are ordered from left to right as $d$ increases). For each model, the table on the right shows the vocabulary size $N$, the embedding dimension $d$, and the value of $m$ where our bound falls below 0.99. These values for more models are shown in Appendix C.

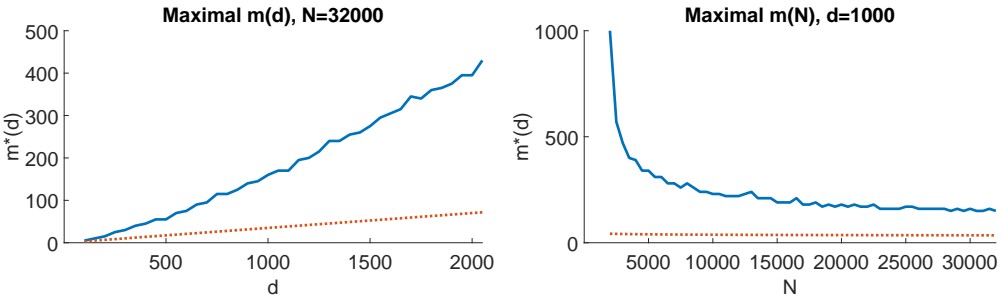

Figure 3: Top-$m$ achieved by random matrices of size $32000 \times d$ with $100 \leq d \leq 2050$ (left) and matrices of size $N \times 1000$ with $2000 \leq N \leq 32000$ (right). The solid blue curves show the values of $m$ for which 0.99 probability is achieved, computed via linear-programming simulation. Our bound is shown by the dotted red curves.

have analyzed the statistical regularities in embeddings and OPMs that emerge through training. In Appendix E, we explicitly compare first and second-order properties of trained and random matrices.

# 7 Conclusions

The softmax bottleneck means that LLMs can only represent a measure-zero subset of the set of all probability distributions for the next token. However, we have shown, both theoretically and empirically, that reasonably sized models can represent the exact probabilities of a large number of the most probable tokens. This calls into question the degree to which the softmax bottleneck imposes limitations on LLMs in most realistic settings.

# 8 Acknowledgements

RB was supported in part by the Israeli Council for Higher Education (CHE) via the Weizmann Data Science Research Center, by the MBZUAI-WIS Joint Program for Artificial Intelligence Research, and by research grants from the Estates of Tully and Michele Plesser and the Anita James Rosen and Harry Schutzman Foundations. DJ was supported by a National Science Foundation (NSF) grant #2213335. We thank an anonymous reviewer for suggesting the simplified proof of Proposition 3 used in this paper.

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

# Appendix

## A   Cumulative probabilities of most likely tokens

Figure 4 shows the average cumulative probabilities of the $m$ most likely tokens in ten stories from the New York Times computed using GPT2. Roughly 200 tokens are needed to predict the next token with 0.9 probability. A threshold of 0.9 is commonly used for nucleus sampling. We note that larger models such as Llama2-7B can empirically produce precise probabilities for about $m = 1000$ tokens.

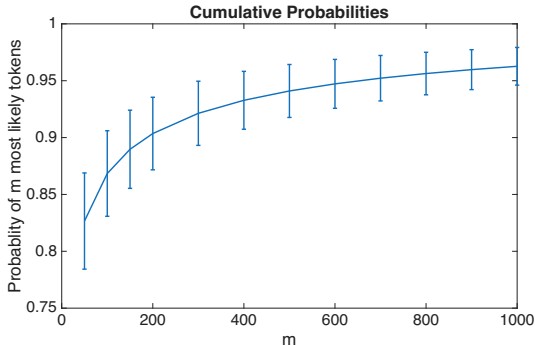

Figure 4: For different values of $m$ we determine the average cumulative probabilities of the $m$ most likely tokens. This is averaged over all embeddings produced by ten stories from the New York Times, across different subject areas. Probabilities are produced by analyzing these stories using GPT2. Error bars show the standard deviation of average values produced by the ten different stories. Note that we have previously shown that even small models, such as Llama2-7B, can empirically produce precise probabilities for about $m = 1000$ tokens.

## B   Independence assumption

Figure 5 compares the bound derived in Equation (4) to a numerical simulation. The probabilities obtained with our bound are slightly smaller than those observed in the simulation due to the statistical dependence of the factors in Equation (6).

## C   Bounds for additional models

We include the lower bound results in Equation (4) for additional models in Table 1.

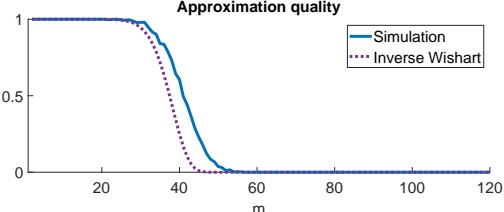

Figure 5: For a random matrix of the size of the GPT2 OPM ($50257 \times 768$), the figure shows top-$m$ probability for the optimal embedding in the row space of $A_m$ obtained with a simulation (solid blue) compared to our inverse-Wishart-based bound (dashed purple). Our bound slightly undershoots the simulation. (A 0.99 probability is achieved at $m = 27$ with the closed form expression compared to $m = 28$ with the simulation.)

| Model | N | d | m |
|-------|-----|-----|-----|
| GPT2 | 50257 | 768 | 26 |
| BERT-Base | 30522 | 768 | 26 |
| T5 Base | 32128 | 768 | 26 |
| XLNet-Base | 32000 | 768 | 26 |
| BERT-Large | 30522 | 1024 | 35 |
| T5-Large | 32128 | 1024 | 35 |
| XLNet-Large | 32000 | 1024 | 35 |
| GPT2-XL | 50257 | 1600 | 54 |
| Olmo2-1B | 100352 | 2048 | 66 |
| Gamma2-2B | 256000 | 2304 | 70 |
| TinyLLaMA | 32000 | 2048 | 71 |
| Phi3-Mini | 32064 | 3072 | 107 |
| Qwen3 | 152064 | 3584 | 113 |
| LLaMA3-8B | 128256 | 4096 | 131 |
| MPT-7B | 50176 | 4096 | 139 |
| LLaMA2-7B | 32000 | 4096 | 143 |
| Falcon-40B | 65024 | 4544 | 152 |
| LLaMA2-13B | 32000 | 5120 | 179 |
| DeepSeek-V3 | 129280 | 7168 | 230 |
| Falcon-40B | 65024 | 8192 | 274 |
| LLaMA2-70B | 32000 | 8192 | 287 |
| GPT3-175B | 50257 | 12288 | 418 |

Table 1: This table lists a number of models, along with their vocabulary size and embedding size. We use our theoretical results to provide a lower bound on the probability that, for a given $m$, a random model of this size can designate any $m$ tokens as the most likely ones. The last column indicates the value of $m$ for which this probability drops below 0.99.

# D  Using natural language probabilities

Our last experiment compares using the uniform distribution to set the target probabilities in Figure 1 to probabilities drawn from the distribution of natural language. For this experiment, we applied GPT2 to contexts from ten New York Times articles, as in Appendix A. Then, for a specific choice of $m$, we randomly selected a token and extracted the top $m$ probability values for the next token after it. We then used LP to determine whether a model could find an embedding that would predict these probabilities. Note that for the LP, we assign the extracted probabilities to random tokens. Figure 6 shows that the bounds with these probabilities and with probabilities drawn from a uniform distribution are almost identical.

# E  Statistical properties of OPM rows

Many prior works have analyzed the statistical properties of embeddings and OPMs (e.g., [3, 14, 4]). In Figure 7, we explicitly compare the statistics of trained and random matrices.

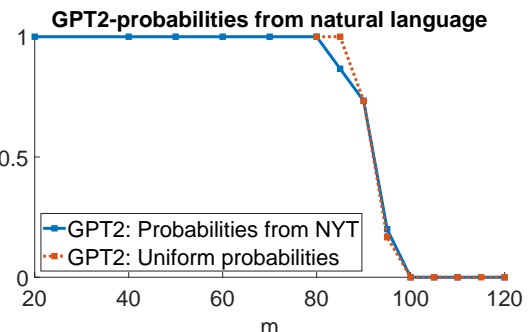

Figure 6: In Figure 1 we showed that GPT2 can produce precise probabilities for $m$ tokens, with $m$ up to about 85 (red curve). These probabilities were drawn from a uniform distribution. In this figure, we add a second version of the experiment (blue curve) that uses probability distributions of next-token probabilities generated by GPT -2 on news articles. We can see there is little difference.

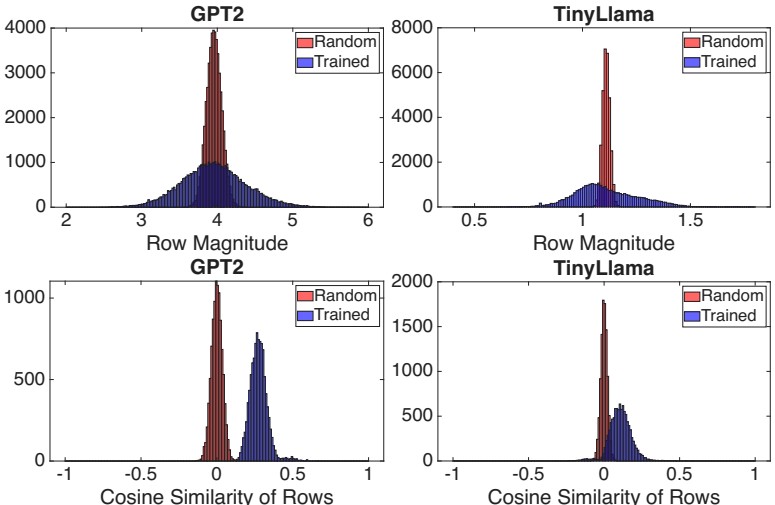

Figure 7: Examining first and second order statistics between rows, we do see a significant difference between random and trained models. In each figure, we compare a trained model to a random model of the same size. In the top row of the figure, we plot a histogram of the magnitudes of the rows. The random matrix is scaled so that its rows have the same mean magnitude as the trained model. We see that the trained models have much greater variance in their magnitude. In the bottom row, we compare the cosine similarity of randomly chosen rows. While the random models have mean-zero cosine similarity, the trained models have almost all positive values, with greater variance. [3] have previously described similar empirical results. Note that GPT-2 is trained with tied weights between the encoding and unencoding matrices, whereas TinyLlama uses untied weights.

