# OpenReview forum: "The Softmax Bottleneck Does Not Limit the Probabilities of the Most Likely Tokens"
_ICLR.cc/2026/Conference — ICLR 2026 Poster_

### Official Review · Reviewer_3zJk · 2025-10-27

**Soundness:** 4
**Presentation:** 3
**Contribution:** 3
**Rating:** 6
**Confidence:** 3

**Summary:**

This paper offers a new look at the so-called "Softmax Bottleneck," which is said to limit the ability of a model as it can struggle to go from inner embeddings to logits over the whole vocabulary: the authors claim that what matters in practice is to predict accurately the probabilities for top tokens, especially given that sampling is usually done from a truncated distribution. The paper first gives theoretical lower bounds showing that indeed any chosen set of m tokens can be the most likely ones, with high probability, and even reach, collectively, a specific probability. It then validates this on GPT‑2 and TinyLlama: in practice this works for somewhat surprisingly larges m.

**Strengths:**

- this is a really inspiring take on the softmax bottleneck, as indeed given the prevalence of truncated sampling, what matters is often the ability to produce valid probabilities for the top tokens of a vocabulary ;
- the authors show the theoretical possibility to produce the embeddings that would result in a given m-subset getting the most probabilities, for  a randomly initialized projection matrix, with some bounds, and even to reach a given probabilities for this subset ;
- the experiments validate their theory for a trained GPT-2 and TinyLlama, far exceeding the lower bounds ;
- the paper is convincingly written, with a well formulated problem, sound proofs and related experiments.

**Weaknesses:**

- the proofs seem sound but this is really to the best of my understanding and I had to trust the authors for this theoretical work. Similarly, it's not clear what to make of such a sentence "since this derivation involves some approximation, we have empirically confirmed that simulations match our theoretical predictions (not shown)";
- although this work proves the existence of "embedding that the OPM will map into the appropriate probability distribution", there is no guarantee that the model will produce it ;
- furthermore, although this is interesting, it does not say anything about the ability of a given network to produce the embeddings for "desired" m-subsets in many contexts. This is not the point of this work, but it does take a practical look at the softmax bottleneck, so my _practical_ concern might be warranted...

**Questions:**

repeating what I see as a weakness:
- what do you think of the reachability question? Your results do establish existence of an embedding $x$ for a given $A$ but I wonder whether a network can be trained to produce that $x$...
- are we sure that same (trained) network would produce the valid embeddings, whatever the context? This seems important to fully grasp the extent of your work.

---

> ### Author Response · Authors · 2025-11-20
> **Addressing feedback**
>
> Thank you for the thoughtful comments.
>
> $\textbf{Empirical confirmation of derivation}$. We have included Figure 6 in the Appendix, which compares our derivation in Theorem 1 with a corresponding simulation. As can be seen, our bound slightly undervalues the top-$m$ probability; for example, it sets the 0.99 probability at $m=27$ in our derivation, instead of $m=28$ in the simulation. We attribute this difference to the assumption of statistical independence.
>
> $\textbf{Will the model produce embeddings for desired sets of $m$ tokens?}$ This is a difficult question to answer, because we do not know what the desired probabilities are, and whether a model is achieving them.  We would point out that the optimal embeddings can be found using linear programming, which is a convex optimization problem.  Furthermore, the embeddings used in our theoretical lower bound are achieved by solving a simple equation operating only in the space of the rows of the OPM that correspond to the top $m$ tokens, so there is reason to believe that these embeddings would be simpler for the model to find.
>
> $\textbf{Effect of context on reachability}$: Our results show that with a suitable $m$, there exist embeddings $x$ that are mapped by both random and trained OPMs to the desired probabilities and that these embeddings can be found either by solving a simple set of equations or by optimization. We therefore expect that a network can be trained to find these embeddings with any context. Our results, however, do not guarantee that a trained network indeed finds these embeddings. If we have misunderstood your question, we apologize and would appreciate it if you could clarify it.

---

> ### Comment · Reviewer_3zJk · 2025-11-25
>
> thank you for these answers. It seems you understood my question about reachability, and that we are left to expect that "a network can be trained to find these embeddings."

---

### Official Review · Reviewer_B1qY · 2025-10-30

**Soundness:** 3
**Presentation:** 3
**Contribution:** 3
**Rating:** 6
**Confidence:** 4

**Summary:**

The work addresses the question of the softmax bottleneck in transformers -- that is, the output projection limiting the production of arbitrary probability distributions. The authors ask whether the softmax bottleneck restricts the LLM from representing the probabilities of the top-m most probable tokens, arguing that the exact probabilities of unlikely tokens are less important. The authors provide theoretical results showing that there exist OPMs that can represent any specific probabilities over top-m tokens for large m and empirical results on GPT-2 and TinyLlama.

**Strengths:**

- Extends on prior work on softmax bottleneck, reframing the problem around top-m probabilities
- The research question is clear as well as the writing
- The derivations give nice lower bounds for random matrices and the theoretical insights are supported by experiments on GPT-2 and TinyLlama.

**Weaknesses:**

- The results show the existence of embeddings that can realize given top-m probabilities, but does not address whether these are learned by real transformers
- Assumes low-probability tokens are irrelevant, but there may be domains (e.g., RL fine-tuning or exploration) where coverage over rare tokens is important

**Questions:**

- What is the role of weight tying here?
- Are there settings (e.g., RL, exploration, or calibration tasks) where representing low-probability tokens accurately would matter, and how would your framework apply there? Do the values of m you found seem sufficient?

---

> ### Author Response · Authors · 2025-11-20
> **Addressing feedback**
>
> Thank you for these thoughtful comments.
>
> $\textbf{Learning embeddings for top-$m$ probabilities}$. This is a difficult question to answer because we do not know what the desired probabilities are or whether a model is achieving them.  We would like to point out that the optimal embeddings can be found using linear programming, which is a convex optimization problem.  Furthermore, the embeddings used in our theoretical lower bound are achieved by solving a simple equation operating only in the space of the rows of the OPM that correspond to the top $m$ tokens, so there is reason to believe that these embeddings would be simpler for the model to find.
>
> $\textbf{Irrelevance of low-probability tokens}$. We agree with reviewers that our claims about the effects of the softmax bottleneck were too broad.  We should consider these effects on inference and learning separately.  We show a new experiment that demonstrates that our results show that trained or random OPMs can capture most or all of the relevant probabilities at inference time (see Figure 7 in the Appendix).  However, reviewers correctly point out that the softmax bottleneck may still limit learning.  We consider it an open question whether the softmax bottleneck limits model training.  We will rewrite the paper to reflect this.
>
> $\textbf{The role of weight tying}$. Weight tying imposes an additional constraint on the output projection matrix (OPM), which can potentially reduce its ability to represent a large number of top tokens. Our experiments, however, suggest that weight tying does not have such a significant effect.  Furthermore, the fact that even random matrices can represent distributions for large $m$ suggests that most trained OPMs would automatically have this property.

---

### Official Review · Reviewer_SQX5 · 2025-10-30

**Soundness:** 1
**Presentation:** 3
**Contribution:** 4
**Rating:** 4
**Confidence:** 5

**Summary:**

This paper tackles the softmax bottleneck problem, i.e. the study of the expressivity of usual neural language models that use a hidden dimension that is smaller than the vocabulary size $N$. In that setup, it has been shown that there exist probability distributions in $\Delta^N$ that cannot be predicted. In this paper, the authors state that language model outputs can match probability distributions with a relatively large support, either by predicting the token set that belong in that support, or even by predicting exactly the probabilities for the tokens in such supports. As a result, they argue that the softmax bottleneck is not a dramatic issue for language models, which they assess through a short experimental analysis.

**Strengths:**

This paper studies a very interesting topic and conducts a novel and relevant theoretical analysis.
- **Theoretical results**: The propositions presented in this paper are novel and provide a more profound understanding of this issue. They shed light on the complexity of reachable distributions, and probabilize (a part of) the set of reachable distributions, which is both a challenging and exciting outcome. Even though the proof in section 4.1 is not perfectly rigorous, it makes very reasonable assumptions and uses the Inverse Wishart distribution in this context, which is a new and very insightful contribution in the context of random matrices for the softmax bottleneck problem. I appreciate that the authors extended their work to fixed top-m probability. Proposition 4 is very elegant. The use of results on the signrank is also very novel and could open new venues for this topic. It leads to a clean result on the minimal dimensions needed to accurately match the supports of probability distributions.

**Weaknesses:**

Although I deeply appreciate the core theoretical results of this work, I quite disagree with some of the claims and conclusions made in the abstract, introduction, and empirical sections. I also believe that this paper could be enhanced by deeper empirical experiments.
- **Claims about broader implications**: It is mentioned in the abstract and introduction that "the softmax bottleneck does not significantly limit the capabilities of LLMs", or that "limitations to the expressiveness of transformers are not really that significant". My understanding of what is proven in the paper is 1- for a single given target probability vector and a random OPM matrix, there exists an $x$ for which the predicted probability matches the top-$m$ target probabilities; 2- if one only cares about the support but wants to match any set of targets on the top-$m$ ranking of tokens, then in most setups $d \approx m/2$ is sufficient. Hence, it is unclear whether matching any set of targets for the correct probabilities (let alone the order of such probabilities) is possible. This is a crucial distinction, as the complexity of matching any permutation of token order is much higher than matching the support as a set. Moreover, even when there would exist output representations that would give the desired probabilities, they might be configured in ways that are very difficult to reach during training, which would limit the applicability of these results to usual training setups. What can thus be concluded from this paper is that the softmax bottleneck phenomenon does not strongly limit the prediction of the next-token probability supports, and that individual "low-entropy" probability vectors should be truthfully matched with non-negligible probability.
- **Lack of clarity in some proofs**: The proof of Proposition 2 is a bit hard to follow and could be made clearer. The first paragraph is a bit confusing and it seems like it could be summarized to convey the idea more directly. Moreover, the statistical independence of the entries of $(A_mA_m^T)^{-1}$ is never verified empirically in the paper.
- **Experimental design**: The experimental section is less appealing than the theoretical section. Figure 1 verifies the lower bound given in Propositions 1 and 2 (and incidentally shows that it is not particularly tight). Figure 2 computes the bound curve for several setups. Figure 3 explores a question that seems loosely related to the topic at hand, and that was covered many times by the anisotropy literature (see works of Ethayarajh et al., among others). A lot of questions remain unanswered, from the necessary experiments to extensions that would have been relevant: given an actual token distribution taken from natural language, how much of the probability supports can a random\trained matrix cover? What are the types of x and s that are observed? In a synthetic controlled setup, e.g. data generated from a bigram with known supports, can a model be trained to properly recover the supports up to $m$ tokens? In cases where the supports are of different sizes, ie where m should be different across target probabilities, what types of solutions are found and with what performance? The experimental section also ignores the second theoretical section of the paper.

In the current state of the paper, the claims and conclusions that are made about the relevance of the softmax bottleneck are not strongly supported by the theoretical section and the experiments. It is particularly important as this article is quite technical, which implies that readers may take these claims for granted without thoroughly reading the paper and understanding its potential limitations. Hence, it is crucial that the claims accurately reflect the results that are presented to avoid any misinterpretation.

Overall, I am excited by the topic and theoretical part (which I would rate 8/10), but I am underwhelmed by the conclusions and experimental sections (which I would rate 2/10).

**Questions:**

- Could you report the results that show independence for the rows of $(A_mA_m^T)^{-1}$?
- By my understanding, Proposition 3 could be proven more easily by setting all a_j at distinct positions on the unit sphere in 2d and setting x_j = a_j?
- Do OPMs actually train to account for larger m values? Is it possible that the learned A_m do not have the used invertibility properties?
- To what extent does token rank and specified ratios would affect the results in section 5?

---

> ### Author Response · Authors · 2025-11-20
> **Addressing feedback**
>
> Thank you for your positive comments on the theory, and for your detailed and constructive comments.
>
> $\textbf{Claims about softmax bottleneck}$: We agree with reviewers that our claims about the effects of the softmax bottleneck were too broad.  We should consider these effects on inference and learning separately.  We show a new experiment that demonstrates that our results show that trained or random OPMs can capture most or all of the relevant probabilities at inference time (see Figure 7 in the Appendix).  However, reviewers correctly point out that the softmax bottleneck may still limit learning.   We will rewrite the paper to reflect this.
>
> $\textbf{Claims about broader implications}$: Our conclusions indeed can be rephrased to better reflect our contributions. Specifically:
> Section 5 shows that for an embedding dimension $d$, there exists an output projection matrix (OPM) that allows assigning the highest probabilities to any subset (= support) of $m \approx d/2$ tokens. Computing such a matrix, however, is generally intractable (NP-hard).  It seems difficult to characterize how this lower bound would change if we restrict the top-$m$ tokens to a specific order or assign them exact probabilities.
>
> Theorems 1-2 and our experiments indicate that with probability near 1 over random matrices, OPMs can express the exact probability ratios for all top-$m$ tokens with fairly large $m$. In an additional experiment that we now include (Figure 5 in the Appendix), we show the value of $m$ as a function of $d$ and $N$. It appears that with large $N$, $m \approx d/5$. Our experiments also indicate that the OPMs in existing LLMs behave similarly to random matrices. We verify this in further experiments with several additional LLMs, including T5-Large, GPT2-XL, and Llama2-7B (see Figure 4 in the Appendix).
>
> In conclusion, learned OPMs can express exact probability ratios for the $m \approx d/5$ most likely tokens.
>
> $\textbf{Clarity of the proof of Theorem 2}$. We thank the reviewer for this suggestion and will post a revised proof shortly.
>
> $\textbf{Statistical independence}$. Figure 6 in the appendix now compares our derived bound with the results of simulations for a random, GPT-sized matrix. Due to the statistical independence assumption, our expression slightly undervalues the corresponding top-$m$ probability. In particular, a 0.99 probability is achieved at $m=27$ using our closed-form expression, compared to $m=28$ using the simulation.
>
> $\textbf{Figure 3– relation to Ethayarajh et al.}$  We will add more discussion of prior work on this topic, and move Figure 3 to the appendix.  We agree this should not be presented as a novel result.
>
> $\textbf{Experimental design}$. We agree that the degree to which natural language next-token distributions can be covered is an important question.  This is quite difficult, though, due to the complexity of this distribution.  Using a bigram distribution is an interesting suggestion, but we expect it to qualitatively differ from the true distribution of natural language, with much higher entropy requiring larger $m$.
>
> $\textbf{Relation between x and s; variation in m}$: Our construction suggests that exactly representing $m$ probabilities requires embeddings with larger scale as $m$ becomes larger.  In a new experiment using 10 news stories and thousands of tokens we find there is some correlation (0.2) between the size of $m$ needed to capture 90% of the probability and the magnitude of the embedding.  This is intriguing but far from definitive.
>
> $\textbf{Necessity of the invertibility property}$. Removing the invertibility requirement of $A_m$ should not generally increase $m$, as this would introduce coupling between logits.

---

> > ### Comment · Reviewer_SQX5 · 2025-11-26
> > **Response to rebuttal**
> >
> > I appreciate that the authors rephrased some key claims in the introduction and conclusion. The new experiments are relevant and interesting. Some concerns remain:
> > - The empirical values of $m$ are derived from cases where the distributions are themselves derived from uniformly sampled weights. As I indicated in my rebuttal, this does not mimic a language modeling setup where the distributions are more sparse and it would be interesting to pick target distributions that are more similar to natural language. The bigram (or n-gram) setup is just one possible setup to achieve this.
> > - I must admit that what is shown in Figure 6 is unclear to me, and the new experimental results could largely benefit from more thorough explanations and a better flow with the rest of the paper.
> > - It is not clear to me how the assumption about the independence of the rows of $(A_mA_m^T)^{-1}$ is verified in random matrices and in learned OPMs with the new experiments.
> >
> > I update my overall score and soundness score to reflect the update about the claims, but I believe that this paper could be of much higher quality and impact if more care and time was given to experimental sections.

---

> > > ### Author Response · Authors · 2025-12-02
> > >
> > > We thank the reviewer for these additional comments and for raising their scores.
> > >
> > > $\textbf{Target distribution.}$ The reviewer pointed out that we measure the ability of a model to produce precise probabilities using probabilities drawn from a uniform random distribution.  We have repeated these experiments using probabilities that GPT2 produces on tokens from New York Times articles.  The results are shown now in FIgure 8, and we can see that the two distributions produce almost identical results.
> > >
> > > $\textbf{Figure 6.}$ We apologize for the lack of clarity with regard to our new experiments.  In these experiments, using New York Times articles, we calculate the next-token probabilities using GPT2.  Given a random token from one of ten New York Times articles, we determine, for varying values of $m$, the cumulative probability of the top $m$ tokens.  We can see that for $m = 200$, this average probability begins to exceed 0.9.  This suggests that at inference time, realistically sized LLMs can capture precisely the probabilities that would be used in nucleus sampling.
> > >
> > > $\textbf{Independence assumption.}$ In deriving our theoretical lower bound, we make an independence assumption, allowing us to estimate the probability that a random matrix can specify the top $m$ tokens with highest probability, using an embedding that is simply derived.  This is a bound, because better embeddings might exist.  We are also able to simulate this empirically, by generating random matrices and determining whether the specific embedding generated successfully specifies the top $m$ tokens, a process whose outcome is affected by dependencies in the entries of $(A_mA_m^T)^{-1}$.  We plot the theoretical and empirical curves in Figure 6, and see that they are very close.

---

### Official Review · Reviewer_kpFi · 2025-11-01

**Soundness:** 3
**Presentation:** 3
**Contribution:** 2
**Rating:** 6
**Confidence:** 3

**Summary:**

The paper investigates both theoretically and empirically the impact of the softmax bottleneck on the ability of large language models to correctly represent the probabilities of the m most probable tokens. Their conclusion is that the softmax bottleneck does not seem to "provide any limitations to LLMs in most realistic settings".

**Strengths:**

- An interesting contribution to the "softmax bottleneck" literature that goes beyond previous work (notably Demeter et al. 2020 and Grivas et al. 2022).
- The paper combines theoretical analyses and empirical results.
- The paper is generally clear and well written.

**Weaknesses:**

- The motivation for the research question is not convincing enough. Why is it so important that models can assign the correct probabilities, summing to (almost) 1, to the m best tokens? If it is possible (and it is for fairly large m's), what does it tell us about language models? If it hadn't been possible, why would that have been an issue, beyond very niche scenarios such as choosing a number or a US state at random?
- The discussion in section 6 could be more detailed. In particular, how do your results complement previously published papers that found that the softmax bottleneck *is* an issue for language models (e.g. Parthiban et al. 2021, Godey et al. 2024)?

**Questions:**

Suggestions rather than questions:
- Please double-check that the papers you cite as arXiv papers have not been published (by that I mean "really" published, in the proceedings of a conference or in a journal). Whenever it is the case, the proper publication should be cited, not the arXiv pre-print. There are multiple cases of this issue in your bibliography.
- Although the paper is generally well written, please refrain from using abbreviations such as "WLOG" (used twice).

---

> ### Author Response · Authors · 2025-11-20
> **Addressing feedback**
>
> Thank you for these helpful comments.
>
> $\textbf{Motivation}$: As previous works have shown, due to the softmax bottleneck, LLMs cannot express all possible probability distributions for the tokens in the vocabulary. A natural requirement, therefore, is that it could express exact probabilities of the most likely tokens for every context, as these are the most likely to be chosen. The question we address is, how large is the set of most likely tokens whose probabilities can be controlled? Moreover, due to the application of the softmax function, obtaining exact probabilities requires controlling the partition function, which is challenging unless all logits can be expressed explicitly. We therefore chose in Theorem 2 to control the ratios of probabilities for the top-$m$ tokens. Letting these ratios sum to 1 is an arbitrary choice, and Theorem 2 can be generalized to a wide range of constants in [0,1].
> We argue that a large $m$ is important beyond niche scenarios. Learning the correct probabilities for the top tokens is crucial in beam search and in nucleus decoding strategies. Also, obtaining proper ‘creative’ responses from LLMs (ones that are not produced simply by the single most likely next token) greatly depends on the accuracy of the probabilities of the set of most likely tokens.
>
> We agree with reviewers that our claims about the effects of the softmax bottleneck were too broad.  We should consider these effects on inference and learning separately.  We show a new experiment that demonstrates that our results show that trained or random OPMs can capture most or all of the relevant probabilities at inference time (see Figure 7 in the Appendix).  However, reviewers correctly point out that the softmax bottleneck may still limit learning.  Prior work has suggested that breaking the softmax bottleneck can improve model performance, but Parthiban et al., 2021 question this and provide evidence that improved model performance is due to other factors.  We consider it an open question whether the softmax bottleneck limits model training.  We will rewrite the paper to reflect this.
>
> $\textbf{Suggestions}$: Thank you for these suggestions. We will modify  our manuscript accordingly.

---

### Author Response · Authors · 2025-11-24
**Revision**

Dear reviewers,
Thank you again for your thoughtful reviews. We have uploaded a revised version of the paper in response to your suggestions. Specifically, (1) We revised the motivation, (2) added an introductory explanation to our proof of Theorem 2, (3) moved Figure 3 to the appendix, and (4) included additional experiments in the appendix, as we described in our response. Our modifications are marked in color (blue for new text, orange striked over for omitted text). We welcome any further thoughts and comments that you may have.

---

### Author Response · Authors · 2025-12-02
**Note to AC**

We thank the ACs for handling our paper. We would like to summarize the contributions of our work, the key points of the reviewers, and our responses.

$\textbf{Contribution:}$ Our work analyzes the “softmax bottleneck”, which shows that by mapping lower-dimensional embeddings to a higher-dimensional space of logits, transformers lose the ability to represent all possible next token probability distributions.  Our work analyzes this representational loss in a novel way, showing theoretically and experimentally that it $\textit{is}$ possible for a transformer to represent the exact probabilities of the $m$ most likely tokens for relatively large $m$.  This raises questions about whether the softmax bottleneck actually produces significant limitations.

$\textbf{Reviewers' points:}$ The reviewers appreciated our theoretical contributions (“inspiring”, 3zjk; “I am excited by the topic and theoretical part (which I would rate 8/10)”, SQX5). The reviewers made valuable comments about the framing and motivation of our work, with suggestions for clarifying our results.   We took these suggestions to heart and significantly rewrote sections of the paper, as shown in the latest version of the paper.

Reviewer SQX5 initially rated the paper at 4, because of the need for further experiments.  We added experiments, verifying a theoretical result and showing that even small transformers can capture most of the probability produced at inference time. In particular, we added experiments with three additional LLMs, which support our previous results.  And we added a further experiment showing the relationship between $m$, the embedding size, and the vocabulary size.  This result implies that if one wishes an LLM to be able to represent precisely the probabilities of the $m$ most likely tokens, then it will need an embedding size of at least $5m$. The reviewer appreciated these and raised their rating to 6, but suggested that we redo an experiment that shows the value of $m$ for which transformers can capture exact probabilities, using the probability distributions of natural language.  We have added this experiment, although, unfortunately, the reviewer is unable to comment on this addition.

The reviewers also questioned whether embeddings are reachable by real systems.  We point out in our response to Reviewer 3zjk that our empirical bounds are based on finding embeddings by solving a convex optimization problem, and that our theoretical bounds are based on finding embeddings through a much simpler process, making it plausible that transformers can find these solutions.  However, it is difficult to determine empirically whether the probability distributions that transformers produce take full advantage of their potential capabilities.

---

### Meta-Review · Area_Chair_z1gK · 2026-01-06

**Summary:**

This paper provides novel theoretical analysis of the softmax bottleneck, demonstrating that even randomly initialized projection matrices can successfully represent top-m token probabilities for relatively large m values. Reviewers appreciated the theoretical contributions, with one calling the work "inspiring" and rating the theoretical section 8/10, praising the use of Inverse Wishart distributions and signrank bounds as elegant and insightful. While initial concerns were raised about overly broad claims, the authors revised their framing to more carefully distinguish inference-time expressiveness from learning dynamics and added experiments with natural language distributions that support their theoretical findings. However, a significant limitation remains: the experimental section was rated only 2/10 by the primary reviewer, lacking validation on controlled setups and insufficient exploration of learnability. Despite this weakness, the consensus is that the theoretical analysis makes valuable contributions to understanding transformer expressiveness, particularly for practical truncated sampling scenarios. We encourage the authors to substantially strengthen the empirical section in the camera-ready version and explore the learnability question in future work. Based on the strong theoretical contributions, I recommend accepting this submission as a poster.

**Reviewer Concerns:**

Addressed: Authors revised claims to distinguish inference-time expressiveness from learning dynamics, added experiments with GPT2 on New York Times articles (Figure 7), and acknowledged that "the softmax bottleneck may still limit learning."

Outstanding: The existence vs. learnability gap remains unresolved. Three reviewers (SQX5, 3zJk, B1qY) independently noted that proving embeddings exist does not demonstrate they are learnable during training. The experimental section (rated 2/10 by SQX5) lacks validation on natural language distributions with controlled setups like bigram models.

**Reviewer Scores:**

SQX5 (4→6, C5): Raised to 6 after claim revisions but maintains Soundness=1 and notes experimental weaknesses persist

3zJk (6, C3): Would maintain 6; appreciates theory but concerned about reachability question

B1qY (6, C4): Would maintain 6; existence vs. learnability gap unresolved

kpFi (6): Would maintain 6; brief review without detailed engagement

---

### Decision · Program_Chairs · 2026-01-26

Accept (Poster)